# Stereo Visual Odometry Pose Correction through Unsupervised Deep Learning

**DOI:** 10.3390/s21144735

**Published:** 2021-07-11

**Authors:** Sumin Zhang, Shouyi Lu, Rui He, Zhipeng Bao

**Affiliations:** State Key Laboratory of Automotive Simulation and Control, Jilin University, Changchun 130022, China; zhangsumin@jlu.edu.cn (S.Z.); lusy19@mails.jlu.edu.cn (S.L.); baozp19@mails.jlu.edu.cn (Z.B.)

**Keywords:** simultaneous localization and mapping (SLAM), visual odometry (VO), unsupervised deep learning, pose correction

## Abstract

Visual simultaneous localization and mapping (VSLAM) plays a vital role in the field of positioning and navigation. At the heart of VSLAM is visual odometry (VO), which uses continuous images to estimate the camera’s ego-motion. However, due to many assumptions of the classical VO system, robots can hardly operate in challenging environments. To solve this challenge, we combine the multiview geometry constraints of the classical stereo VO system with the robustness of deep learning to present an unsupervised pose correction network for the classical stereo VO system. The pose correction network regresses a pose correction that results in positioning error due to violation of modeling assumptions to make the classical stereo VO positioning more accurate. The pose correction network does not rely on the dataset with ground truth poses for training. The pose correction network also simultaneously generates a depth map and an explainability mask. Extensive experiments on the KITTI dataset show the pose correction network can significantly improve the positioning accuracy of the classical stereo VO system. Notably, the corrected classical stereo VO system’s average absolute trajectory error, average translational relative pose error, and average translational root-mean-square drift on a length of 100–800 m in the KITTI dataset is 13.77 cm, 0.038 m, and 1.08%, respectively. Therefore, the improved stereo VO system has almost reached the state of the art.

## 1. Introduction

Visual simultaneous localization and mapping (VSLAM) is a critical research direction in robot and scene understanding and plays an essential role in the field of positioning and navigation. At the heart of VSLAM is visual odometry (VO), which estimates a camera’s ego-motion using an interframe continuous image. Over the past decade, researchers have done much research on VO systems. Significantly, several state-of-the-art VO systems have been designed based on feature point matching [1,2,3] and constant gray hypothesis [4,5,6].

However, classical VO systems have many environmental assumptions, such as illumination invariance assumption, static scene assumption, and no significant occlusions assumption. Because of these assumptions, many VO systems cannot run in challenging environments. With the increase of large-scale datasets, more and more questions are raised about whether it is possible to understand and tackle the environmental assumptions of classical VO systems from a data-driven method.

Recently, researchers use deep learning (DL) methods to recover camera motion from continuous image frames [7,8,9,10] or predict the camera pose concerning the scene from a single image frame [11,12,13]. These DL-based methods may compensate for the classical VO’s assumptions, thereby being robust to moving objects, uneven illumination, and obvious occlusion. However, most of these methods learn directly from raw images and rarely consider the geometric models of classical VO system, which are considered the basic principle of the VO system and the interpretability and transferability of the classical VO system. To date, the accuracy of the VO system based on end-to-end approaches has not exceeded that of the classical VO system.

In addition, other methods use the DL to enhance the classical VO system. For example, the depth map predicted by the neural network is used to restore the scale of the monocular VO system [14], and the neural network is used to replace the feature extraction and feature matching in the original VO system [15] or is used for loop closure detection in VSLAM system to improve the accuracy of the VO system [16]. By combining DL with the classical VO system, these methods maintain the interpretability and transferability of the classical VO system and use the capacity and flexibility of the data-driven method to improve the robustness and accuracy of the classical VO system.

Most DL-based methods use supervised learning schemes, which require a large number of labeled datasets. However, labeling large amounts of data is time consuming and expensive, which has strong limitations for the model’s training [17]. For the VO system, since limited labeled data cannot train a robust neural network, the robots fail to operate in a new and complex environment. However, the unsupervised learning schemes can make up for this shortcoming, which can improve the performance by increasing the size of datasets without annotated ground truth.

In this work, we do not adopt the solution of completely abandoning the classical VO system and obtaining the whole interframe pose change from the data alone. Instead, we integrate a classical stereo VO system and an unsupervised neural network model. Through the data-driven method, the deep neural network is used to learn a pose correction, which is used to correct the pose of the classical stereo VO system to make it closer to the ground truth (the real pose of the camera). The unsupervised stereo visual odometry pose correction network takes the prior pose produced by classical stereo VO system and stereo color images as input and outputs pose correction, depth map, and explainability mask simultaneously (see Figure 1). The main contributions of this work are summarized as follows:(1)An unsupervised stereo visual odometry pose correction network is used, which can be trained without labeled data.(2)During training, the spatial and temporal properties of the stereo image sequence are used to model the camera ego-motion, and a modified version of the U-Net encoder–decoder [18] is designed.(3)An unsupervised stereo visual odometry pose correction network is used that can output camera pose correction, left–right depth map, and left–right explainability mask simultaneously.(4)Experiments show the stereo visual odometry pose correction network can significantly improve the positioning accuracy of the classical stereo VO system, and the improved stereo VO system has almost reached the state of the art.

The rest of this paper is summarized as follows. Section 2 provides an overview of geometry-based VO, supervised deep learning of VO, unsupervised deep learning of VO, and hybrid VO. The system architecture of the proposed unsupervised stereo visual odometry pose correction network is provided in Section 3. Section 4 shows the training losses. The results of the open datasets are presented in Section 5. Finally, Section 6 concludes the study.

## 2. Related Work

### 2.1. Classical VO

Camera pose estimation is a fundamental and widely studied problem in the field of computer vision. Classical VO systems are mainly based on multiview geometry. Geometry-based VO/SLAM methods are mainly divided into two categories: feature-based methods [2,19,20] and direct methods [5,6]. Feature-based method constructs feature reprojection error by feature matching and then minimizes feature reprojection error to estimate camera pose. ORB-SLAM2 [19] is the most representative feature-based SLAM system that uses oriented fast and rotated brief (ORB) features to match feature points and divides tracking, mapping, and loop closure detection into three parallel threads. Compared to the feature-based methods, the direct methods are based on the assumption of gray invariance, and the camera pose is obtained by minimizing the photometric error of the corresponding pixels of adjacent frames. Direct sparse odometry (DSO) [6] is the most successful direct visual SLAM system that maintains a sliding window and optimizes all the keyframes in the window to obtain the camera pose and map points. Moreover, a semidirect method combines the above two methods, and its representative work is semidirect monocular visual odometry (SVO) [21]. The above methods all have problems with the previously mentioned classical VO systems.

### 2.2. Supervised Deep Learning VO

Supervised VO system uses labeled data to train a deep neural network, and the image input into the network can directly obtain the camera pose. One of the first works in this area was PoseNet proposed by Konda et al. [22]. This approach uses a convolutional neural network (CNN) to prepare a classifier on the image recognition datasets and then uses transfer learning to train a pose estimator, which estimates the camera’s six-DoF pose. Li et al. [23] then extended PoseNet to present a dual-stream CNN to achieve indoor relocalization in challenging environments. Walch et al. [24] combined the CNN and the long short-term memory neural network (LSTM) to regress the camera pose for indoor and outdoor scenes. Kendall and Cipolla [25] used a Bayesian convolutional neural network to regress the six-DoF camera pose from a single RGB image. Clark et al. [26] proposed to use a CNN–recurrent neural network(RNN) model to regress the camera pose from the monocular image sequence. Muhamad et al. [27] applied curriculum learning to the geometric problem of the monocular VO system and proposed a geometry-aware objective function to regress the six-DoF camera pose. Wang et al. [28] proposed the DeepVO, which utilizes a combination of CNN and RNN to estimate directly poses from the raw RGB image. This approach uses CNN to learn geometric feature representation and uses RNN to learn the association between image sequences.

### 2.3. Unsupervised Deep Learning VO

The main reason for restricting the development of the supervised VO system is that it requires tens of thousands of labeled data to train the network. Therefore, researchers are increasingly interested in the unsupervised VO system that does not require a ground truth label. The SfM-Learner proposed by Zhou et al. [7] is the first unsupervised VO system that jointly estimates camera pose and depth map. Bian et al. [10] extended SfM-Learner to propose a geometry consistency loss and an induced self-discovered mask to solve the scale inconsistent issue in SfM-Learner. Barnes et al. [29] proposed an unsupervised approach to ignore “distractors” in-camera images, which makes the vehicle motion estimation more accurate in the cluttered urban environment. Yin et al. [9] extracted the geometric relationships from the prediction of each module of the neural network output and then combined them as image reconstruction losses, reasoning about static and dynamic scene parts, respectively. Zhao et al. [30] combined 2D optical flow and depth map of the monocular image to generate 3D dense optical flow; then, based on 3D dense optical flow, they achieved six-DoF relative pose estimation. Li et al. [17] proposed the DeepSLAM, which uses a deep recurrent convolutional neural network (RCNN) to simultaneously generate pose estimate, depth map, and outlier rejection mask. Zhang et al. [31] presented a monocular VO system that combines the geometry-based method and the unsupervised deep learning. Liu et al. [32] presented a deep-learning-based RGB-D visual odometry system, which takes RGB image and depth image as input and outputs camera pose through a dual-stream structure of a recurrent convolutional neural network.

### 2.4. Hybrid VO

The above learning-based methods have a common problem: they do not consider multiview geometry constraints of the classical VO system when constructing the VO system. In order to address this problem, researchers combine learning-based methods and classical VO systems to achieve better results. Valente et al. [33] proposed to use a CNN model to fuse the pose estimation results of 2D laser scanners and monocular cameras for odometry estimation. Yang et al. [34] proposed a novel monocular visual odometery framework, which uses deep learning methods to predict depth, pose, and uncertainty. Then, the predicted attributes are applied to the front-end tracking and the back-end non-linear optimization of the DSO. Tateno et al. [14] used CNN to estimate the dense depth map and then used the depth map to optimize the map points estimated by the monocular SLAM. Sarlin et al. [35] proposed a graph neural network with an attention mechanism to match between two sets of local features. Ji et al. [36] fused the dense depth map generated by the CNN and a sparse map generated by the feature-based SLAM to generate a dense monocular reconstruction. Rico et al. combined camera motion model and deep neural network via particle filter to improve the accuracy and robustness of the VO system.

In summary, unsupervised deep learning and hybrid VO technology are promising new research trends in the field of visual odometry research. Our work belongs to hybrid VO, where we combine the unsupervised deep learning’s ability to use unlabeled sensor data with the multi-view geometry constraints of classical VO systems to improve VO’s performance further.

## 3. System Overview

According to the testing framework in Figure 1, the trained stereo visual odometry pose correction network can be viewed as the back end of the stereo visual odometry to correct the estimated camera pose of the classical stereo VO system and generate a more accurate camera pose. At the same time, the stereo depth map and the stereo explainability mask are also generated.

The training scheme of the stereo visual odometry pose correction network is shown in Figure 2. The network structure is a symmetrical structure from top to bottom, and each part of the top and bottom is a modified version of the U-Net encoder–decoder [18]. The network inputs are two pairs of images (source image and target image) of the stereo camera. After each pair of images passes through the encoder network and the decoder network in turn, the decoder network generates the corresponding depth map and explainability mask. After dimensionality reduction through the fully connected layer, the upper and lower parts of the network are connected with the prior pose generated by the classical stereo VO system. Then, the pose correction value is obtained after dimensionality reduction through the fully connected layer. In the network propagation, we use unconstrained 
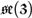
 Lie algebra, $\xi_{t+1,t}^{corr}\in R^{6\times 1}$ to parameterize the correction. When the network outputs the final correction, we use the exponential map to generate an SE(3) correction as follows:(1)$$T_{t+1,t}^{corr}=Exp\left(\xi_{t+1,t}^{corr}\right)$$
where $\xi_{t+1,t}^{corr}$ is the Lie algebra form of the pose correction value from *k*th frame to (k+1)th frame, Exp((·) is the exponential map. Therefore, we can use $T_{t+1,t}^{corr}$ to correct a classical VO estimate $T_{t+1,t}^{vo}$ to obtain the accurate pose $T_{t+1,t}^{*}$. The correction formula is as follows:(2)$$T_{t+1,t}^{*}=T_{t+1,t}^{corr}T_{t+1,t}^{vo}$$

The whole network consists of an encoder network, decoder network, and a fully connected network. The encoder network comprises five parts, and each part comprises a 2D convolution layer with stride 2, a ReLu activation layer, and a batch normalization layer. We do not use the pooling layer after the convolution layer because the pooling layer will enhance the invariance of image features, which is harmful to the VO system. We divide the network into three subnetworks at the bottleneck: depth estimation subnetwork, explainability mask subnetwork, and pose correction subnetwork. For the depth estimation subnetwork, we use the 2D-transposed convolution layer [37] with the ReLu activation layer for upsampling, the 2D convolution layer with stride 1 for depth prediction, and the skip connection to prevent gradient explosion and gradient disappearance caused by the network being too deep. For the explainability mask subnetwork, we use 2D-transposed convolution layer with the ReLu activation layer for upsampling, but the final 2D-transposed convolution layer with a sigmoid activation layer, which compresses the pixel values to be within (0,1). We use the fully connected layer for the pose correction subnetwork to reduce and weigh the features extracted from the encoder network.

We establish the loss functions through the spatial and temporal geometric consistency of stereo image sequences. In spatial geometric constraints, we warp a left (right) image into a right (left) image and evaluate photometric reconstruction loss and disparity loss according to the source image and the composite image. In temporal geometric constraints, we warp a source image into a target image and evaluate photometric reconstruction loss and explainability mask loss according to the source image and the target image. Using these loss functions and minimizing them all together, the network uses unsupervised learning to estimate the pose correction, the stereo depth map, and the stereo explainability mask.

## 4. Loss Function

This section introduces the loss functions developed to train the stereo visual odometry pose correction network. In order to improve the constraint of the loss function, we establish the loss function from the spatial and temporal geometric consistency of the stereo image sequences. We use the stereo color image as input. The spatial geometric consistency refers to the projection constraint between the pixels corresponding to the same world point in the left and right images at the same time. The temporal geometric consistency is the projective constraint among monocular images corresponding to the same world pixel. The establishment of these two losses is shown in Figure 3.

### 4.1. Spatial Image Loss of Left–Right Image Pairs

The spatial image loss function is constructed by the left–right image pairs’ geometric constraints, which mainly include the photometric consistency loss and the disparity consistency loss of the left–right image pairs.

#### 4.1.1. Photometric Consistency Loss

The photometric consistency loss of left–right image pairs refers to the projection pixel error of the left–right image pairs. For a left–right image pair, the overlapping area of the images is the area where the reprojection points are located. In this area, each pixel can find the corresponding pixel in another image. After the distortion correction of the stereo image, the two corresponding pixel points should be on the same horizontal line. Assuming that the distance between the two pixel points is $D_{p}$, the pixel point coordinate of the left image is $p_{l}\left(u_{l},v_{l}\right)$, and the corresponding pixel point coordinate of the right image is $p_{r}\left(u_{r},v_{r}\right)$, we can obtain the following geometric constraints:(3)$$\begin{array}{cc} \; & u_{l}=u_{r}+D_{p}\\ \; & v_{l}=v_{r} \end{array}$$
where $D_{p}$ is the parallax. Given the pixel point’s depth value $D_{dep}$, the parallax $D_{p}$ can be calculated by
(4)$$D_{p}=\frac{Bf}{D_{dep}}$$
where *B* is the baseline of the stereo camera and *f* is the focal length. Therefore, according to the depth map ${\hat{D}}_{dep}$ output from the stereo visual odometry pose correction network, we can obtain the parallax map ${\hat{D}}_{p}$ corresponding to the left and right images, respectively. Based on ${\hat{D}}_{p}$, we can warp the left (right) image into a composite image corresponding to the right (left) image through spatial transformer [38]. Based on the source image and composite image, the left–right photometric consistency losses is defined as follows:(5)$$\begin{array}{cc} \; & L^{l}=\lambda_{s}L^{SSIM}\left(I_{l},I_{l}^{*}\right)+\left(1-\lambda_{s}\right)L^{l_{1}}\left(I_{l},I_{l}^{*}\right)\\ \; & L^{r}=\lambda_{s}L^{SSIM}\left(I_{r},I_{r}^{*}\right)+\left(1-\lambda_{s}\right)L^{l_{1}}\left(I_{r},I_{r}^{*}\right) \end{array}$$
where $I_{l}^{*}$ and $I_{r}^{*}$ are the composite left and right images from the source right image $I_{r}$ and source left image $I_{l}$, respectively. $L^{l_{1}}$ is the L1 norm, $L^{SSIM}$ is the structural similarity (SSIM) metric [39], $\lambda_{s}$ is the weight of weighing the SSIM loss and the L1 loss, which is obtained through network learning. $L^{SSIM}\left(I,I^{*}\right)$ is defined as follows:(6)$$\begin{array}{cc} \; & L^{SSIM}\left(I,I^{*}\right)=\frac{1-SSIM(I,I^{*})}{2}\\ \; & SSIM\left(I,I^{*}\right)=\frac{\left(2\mu_{I}\mu_{I^{*}}+c_{1}\right)\left(2\sigma_{II^{*}}+c_{2}\right)}{\left(\mu_{I}^{2}+\mu_{I^{*}}^{2}+c_{1}\right)\left(\sigma_{I}^{2}+\sigma_{I^{*}}^{2}+c_{2}\right)} \end{array}$$
where *I* is the source image, $I^{*}$ is the composite image, $\mu_{I}$ is the source image’s mean, $\mu_{I^{*}}$ is the composite image’s mean, $\sigma_{I}$ is the source image’s variance, $\sigma_{I^{*}}$ is the composite image’s variance, $\sigma_{II^{*}}$ is the covariance of the source image and the composite image, and $c_{1}$ and $c_{2}$ are constant.

#### 4.1.2. Disparity Consistency Loss

Based on the parallax map and image width, the disparity map is defined as follows:(7)$$D_{disp}={\hat{D}}_{p}\times \omega$$
where ω is the image width. It can be seen from the above formula that the disparity map of the left and right image is constrained by ${\hat{D}}_{p}$. Therefore, we can use $D_{p}$, $D_{disp}^{r}$, and $D_{disp}^{l}$ to synthesize $D_{disp}^{l^{*}}$ and $D_{disp}^{r^{*}}$. Based on source disparity maps and composite disparity maps, the disparity consistency losses is defined as follows:(8)$$\begin{array}{cc} \; & L_{disp}^{l}=L^{l_{1}}\left(D_{disp}^{l},D_{disp}^{l^{*}}\right)\\ \; & L_{disp}^{r}=L^{l_{1}}\left(D_{disp}^{r},D_{disp}^{r^{*}}\right) \end{array}$$
where $D_{disp}^{l}$ and $D_{disp}^{r}$ are the source left and right disparity maps, respectively. $D_{disp}^{l^{*}}$ and $D_{disp}^{r^{*}}$ are the composite left and right disparity maps, respectively. $L^{l_{1}}$ is the L1 norm.

### 4.2. Temporal Image Loss of a Sequence of Monocular Imagery

The temporal image loss function is constructed by the geometric projective constraint of the corresponding points in two consecutive monocular images, which mainly include the photometric consistency loss and the explainability mask loss of two consecutive monocular images.

#### 4.2.1. Photometric Consistency Loss

Unlike the photometric consistency loss in the previous section, which mainly focuses on the spatial information of left–right images at the same time. The photometric consistency loss here focuses on the temporal information between consecutive images of a monocular camera. We use the depth map ${\hat{D}}_{dep}$ of the stereo visual odometry pose correction network output, the camera intrinsics, and the corrected pose change between frames to construct an inverse compositional warping function. Assuming $I_{k}$ and $I_{k+1}$ are the *k*th and (k+1)th frame, $p_{k}\left(u_{k},v_{k}\right)$ is a pixel point in the frame $I_{k}$, $p_{k+1}\left(u_{k+1},v_{k+1}\right)$ is the corresponding pixel point in the frame $I_{k+1}$. We can obtain the conversion relationship between the $p_{k}$ and $p_{k+1}$ through the multiview geometry method. First, we compute $p_{k}$ correspond to the 3D point $p_{3D_{k}}={\left[x_{k}\;y_{k}\;z_{k}\right]}^{T}$ in the scene
(9)$$p_{3D_{k}}=D_{dep_{k}}\left[\frac{u_{k}-c_{u}}{f_{u}}\;\frac{v_{k}-c_{v}}{f_{v}}\;1\right]$$
where $D_{dep_{k}}$ is the depth value of pixel point $p_{k}$, $\left(c_{u},c_{v}\right)$ is the camera’s principal point, and $\left(f_{u},f_{v}\right)$ are the camera focal lengths in the horizontal and vertical directions, respectively. Second, we use the corrected pose change between $I_{k}$ and $I_{k+1}$ to transform $p_{3D_{k}}$ to its 3D position in the $I_{k+1}$,
(10)$$p_{3D_{k+1}}=T_{t+1,t}^{*}p_{3D_{k}}$$

Finally, The 3D coordinates $p_{3D_{k+1}}$ are reprojected into the (k+1)th frame $I_{k+1}$ to obtain $p_{k+1}$
(11)$${\left[p_{k+1}^{T}\;1\right]}^{T}=\left[\begin{array}{ccc} f_{u} & 0 & c_{u}\\ 0 & f_{v} & c_{v}\\ 0 & 0 & 1 \end{array}\right]\frac{1}{z_{k+1}}p_{3D_{k+1}}$$

According to the above transformation, we use the original pixel intensity of the source image to fill the predicted pixel position to construct the target image. As in the previous section, we use the spatial transformer to perform differential image warping. Therefore, the photometric consistency losses of the consecutive images of a monocular camera can be constructed as follows:(12)$$\begin{array}{cc} \; & L_{k,k+1}=W_{k}\left(\lambda_{s}L^{SSIM}\left(I_{k},I_{k}^{*}\right)+\left(1-\lambda_{s}\right)L^{l_{1}}\left(I_{k},I_{k}^{*}\right)\right)\\ \; & L_{k+1,k}=W_{k+1}\left(\lambda_{s}L^{SSIM}\left(I_{k+1},I_{k+1}^{*}\right)+\left(1-\lambda_{s}\right)L^{l_{1}}\left(I_{k+1},I_{k+1}^{*}\right)\right) \end{array}$$
where $I_{k}^{*}$ is the synthesized image from the (k+1)th frame $I_{k+1}$, $I_{k+1}^{*}$ is the composite image from the *k*th frame $I_{k}$, and $W_{k}$ and $W_{k+1}$ are the explainability mask of the corresponding *k*th frame and (k+1)th frame, respectively. We will discuss the mask in the next section.

#### 4.2.2. Explainability Mask Loss

In the actual operating environment of the camera, dynamic objects will cause significant errors in photometric loss and geometric loss. In order to solve this, we introduce an explainability prediction network, which outputs a weight *W* for each pixel of each target–source pair, which reflects the possibility of dynamic objects. Each pixel loss is weighted by the explainability mask, W∈(0,1). To prevent all explainability weights from being zero, we use the cross-entropy loss to establish a regularization term $L_{exp}$, which can ensure each pixel has a constant label 1. The definition of $L_{exp}$ is as follows:(13)$$L_{exp}=-logW$$

In summary, our overall loss function is expressed as
(14)$$\begin{array}{cc} L_{total}= & \alpha_{1}\left(L^{l}+L^{r}\right)+\alpha_{2}\left(L_{disp}^{l}+L_{disp}^{r}\right)+\\ \; & \alpha_{3}\left(L_{k,k+1}^{l}+L_{k+1,k}^{l}+L_{k,k+1}^{r}+L_{k+1,k}^{r}\right)+\alpha_{4}L_{exp} \end{array}$$
where $\alpha_{1},\alpha_{2},\alpha_{3},\alpha_{4}$ are hyperparameters.

## 5. Experimental Evaluation

In this section, we will extensively evaluate our method on the KITTI dataset [40]. We compare our system with other excellent VO systems. We use the KITTI odometry sequences 00, 02, 05-10, and 24 training sequences from the “city”, “residential”, and “road” categories in the raw KITTI dataset as our training datasets. The training dataset contains approximately 46,000 training pairs. As shown in Figure 4, the training datasets include many challenging scenes, such as moving objects, uneven illumination, evident occlusion, etc. Through training in these challenging scenes, the stereo visual odometry pose correction network has better robustness.

### 5.1. Implementation Details

We use the DL framework PyTorch [41] to train the stereo visual odometry pose correction network. Moreover, all the training uses Adam optimizer [42] for 30 epochs. We preprocessed images before training: all images are resized to 240×376 pixel and whitened using the ImageNet [43] statistics. The learning rate is set to $6\times 10^{-3}$. We reduce the learning rate by a factor of 0.5 every five epochs. All fully connected layers use hyperparameters with a dropout of 0.5 and a weight decay coefficient of $4\times 10^{-6}$. The loss weightings are $\left[\alpha_{1},\alpha_{2},\alpha_{3},\alpha_{4}\right]=\left[1,1,1,0.08\right]$. In the process of network training, we use all datasets except the test sequence as the training set to train the stereo visual odometry pose correction network. After training, we evaluate our network on the test sequence.

### 5.2. Visual Odometry Evaluation

Our network can match any classical stereo VO system, and the specific stereo visual odometry pose correction network for the stereo VO system can be generated after training. Herein, we train our system with the stereo VO system libviso2-s [20] and show that our approach improves the localization accuracy of this stereo VO system. Our experiments are divided into two parts. First, we evaluate the improvement degree of the classical stereo VO system by the stereo visual odometry pose correction network, and then we compare the corrected classical stereo VO system with other state-of-the-art VO systems.

#### 5.2.1. Evaluation Metrics

To evaluate the improvement degree of classical stereo VO system by the stereo visual odometry pose correction network, we use two error metrics: cumulative absolute trajectory error and mean segment error. These two metrics are defined as follows:

*(1) Cumulative Absolute Trajectory Error (c-ATE):* Cumulative absolute trajectory error is the sum of the rotational or translational differences between the poses estimated by the VO system and the ground truth poses. *c-ATE* is little affected by good trajectory overlaps so that it can show clear trends. However, the poor (but isolated) relative transforms will cause it to produce a large error. Concretely, $e_{m-ATE}$ is defined as follows:(15)$$e_{c-ATE}=\sum_{p=1}^{q}∥ln{\left({\hat{T}}_{p,0}^{-1}T_{p,0}\right)}^{\vee }∥$$
where the notation $ln{\left(\cdot \right)}^{\vee }$ returns rotational or translational components depending on context.

*(2) Segment Error:* There are two steps to receive the segment error. The first step is to obtain the average of the end-point error of all given segments within the trajectory, and the second step is to normalize the average according to the segment length. In contrast to cumulative absolute trajectory error, since segment error is calculated from multiple starting points within the trajectory, segment error has good robustness to the isolated degradations. Concretely, $e_{seg}\left(s\right)$ is defined as
(16)$$e_{seg}=\frac{1}{sN_{s}}\sum_{p=1}^{N_{s}}∥ln{\left({\hat{T}}_{p+s_{p},p}^{-1}T_{p+s_{p},p}\right)}^{\vee }∥$$
where *s* is the segment length, $N_{s}$ is the number of segments of the given length, and $s_{p}$ is the number of poses in each segment. In this work, we calculate all segment errors when s∈[100,200,300,⋯,800](m).

To evaluate the corrected classical visual odometry, we adopt more evaluation metrics, including the average translational error $t_{err}\left(\%\right)$ and rotational errors $r_{err}{(}^{\circ }/100\;m)$ of the subsequences of length (100, 200, ..., 800) meters, absolute trajectory error, and relative pose error. They are defined as follows:

*(1) Absolute Trajectory Error (ATE):* Absolute trajectory error measures the root mean squared error between predicted camera poses [xyz] and ground truth. ATE can well evaluate the global consistency of the estimated trajectory. However, ATE only considers the translational errors. Concretely, $e_{ATE}$ is defined as
(17)$$e_{ATE}={\left(\frac{1}{q}\sum_{p=1}^{q}{∥trans(Q_{p}^{-1}SP_{p})∥}^{2}\right)}^{\frac{1}{2}}$$
where $Q_{1:q}$ is the ground truth trajectory, $P_{1:q}$ is the estimated trajectory, *S* is a transformation matrix that transforms the estimated trajectory and the ground truth trajectory into the same coordinate system, and trans(·) is the translation components of the absolute trajectory error.

*(2) Relative Pose Error (RPE):* Relative pose error measures frame-to-frame relative pose error. Compared with ATE, RPE considers both translational and rotational errors. The relative pose error at time step *i* as
(18)$$E_{i}={\left(Q_{i}^{-1}Q_{i+\Delta }\right)}^{-1}\left(P_{i}^{-1}P_{i+\Delta }\right)$$
where Δ is the fixed time interval between two frames, we obtain the $e_{RPE}$ by computing the root mean squared error over all time indices of the translational and rotational component. Concretely, $e_{RPE}$ is defined as
(19)$$\begin{array}{cc} \; & e_{RPE_{trans}}={\left(\frac{1}{q}\sum_{p=1}^{q}{∥trans(E_{i})∥}^{2}\right)}^{\frac{1}{2}}\\ \; & e_{RPE_{rot}}={\left(\frac{1}{q}\sum_{p=1}^{q}{∥rot(E_{i})∥}^{2}\right)}^{\frac{1}{2}} \end{array}$$
where rot(·) is the rotation components of the relative pose error.

#### 5.2.2. Improvement Degree of Classical Stereo VO System

Figure 5 shows the north–east projection of each trajectory. As shown in Figure 5, the corrected trajectories appear to be significantly more accurate than the original libviso2-m estimate, almost coinciding with the ground truth trajectories. Especially in sequence 00, the improvement effect is the most obvious. The original trajectory has a large deviation from the ground truth trajectory, but the corrected trajectory almost coincides with the ground truth trajectory after correction. Figure 6 plots c−ATE and mean segment errors for test sequences 00, 02, 05-10. As shown in the figure, whether in c−ATE or segment error, the corrected trajectories have been greatly improved, compared to the original trajectories, especially in the translational error in the segment error, the original trajectories error is showing an increasing trend. In contrast, the corrected trajectories error appears a downward trend, which shows that the corrected trajectories are moving closer and closer to the ground truth trajectories in terms of translation. In the rotational error in the segment error, we have also achieved good results, and the rotational error is almost zero at the end of the corrected trajectory. However, the corrected trajectory does not perform as well as the original trajectory at the end of sequence 06. We suspect the effect results from large rotating, which is challenging to construct an accurate image through photometric consistency. In the cumulative error, the green line is always below the orange line, and the growth trend is much smaller than the orange line, which shows that our network significantly reduces the cumulative error of the VO system. In summary, it is believed that our stereo visual odometry pose correction network plays an essential role in improving the accuracy of pose estimation for the classical VO system.

#### 5.2.3. Corrected Classical Stereo VO System Evaluation

We compare the corrected pose with pure deep learning methods [7,8,10,44,45] (SfM-Learner, Depth-VO-Feat, SC-SfMLearner, ss-DPC-Net, and ESP-VO), geometry-based methods including DSO [6], libviso2-s [20], ORB-SLAM2 [19] (with loop closure), and CNN-SVO [46]. We use the image’s original size as the input of the geometry-based methods. Table 1 shows the result for test sequences 00, 02, 05-10 of the KITTI dataset. As shown in the table, the corrected pose outperforms the pure deep learning methods in tracking accuracy. Compared with ESP-VO, since our network uses an unsupervised training method without labeled data, we can use more datasets (raw KIITI data) training networks to make the network more robust. However, ESP-VO uses the supervision method to train networks, which can only use labeled data, and therefore, the datasets for training the network are limited. It can be seen from the results that the unsupervised learning method benefits from using more datasets for training. At the same time, compared with other unsupervised learning methods, we have achieved better results because we adopt more constrained loss functions and retain the multi-view geometric constraints of the classical stereo VO system. Compared to geometry-based methods, the corrected pose outperforms the libviso2-s based on the feature-based method, DSO based on the direct method, and CNN-SVO based on the hybrid method. Compared to ORB-SLAM2, the corrected pose shows less translation drift $t_{err}$, absolute trajectory error ATE, and the translation components of the relative pose error RPE(M), which indicates that the stereo visual odometry pose correction network significantly improves the translation accuracy of classical VO system. Although ORB-SLAM2 shows less rotation drift, the corrected pose is very close to ORB-SLAM2 in the rotational aspect.

At the same time, from this table, we can infer how much libviso2-s is improved by the stereo visual odometry pose correction network. The original libviso2-s is a general stereo VO system, and its positioning accuracy is not as good as CNN-SVO and ORB-SLAM2. After correction by the stereo visual odometry pose correction network, the positioning accuracy of libviso2-s is better than CNN-SVO and also comparable to ORB-SLAM2 (ORB-SLAM2 is considered as the VO system with the highest positioning accuracy so far). Because the stereo visual odometry pose correction network can match all stereo visual odometry, in this study, we only match with libviso2-s. If the network matches with the state-of-the-art stereo VO system, its positioning will be more accurate. In summary, the stereo visual odometry pose correction network can significantly improve the positioning accuracy of the classical VO system, and the improved stereo VO system has almost reached the state of the art.

In addition, we evaluate the time complexity of the algorithm. We test the corrected classical stereo VO system (libviso2-s + stereo visual odometry pose correction network), SfMLearner, and ORB-SLAM2 on a computer with an Nvidia GeForce RTX 3070 with 8 GB memory. We find that the stereo visual odometry pose correction network only needs 9.53 ms for each camera pose correction, and the corrected libviso2-s only needs 42.73 ms for each camera pose estimation. ORB-SLAM2 needs 37.2 ms for each camera pose estimation. Although the corrected libviso2-s takes 5.53 ms longer than ORB-SLAM2 for each camera pose estimation, the corrected libviso2-s achieves higher accuracy and can keep running in real-time at 43.73 ms. However, SfMLearner, an unsupervised deep learning VO, needs 83.26 ms for each camera pose estimation. It can be inferred that our hybrid VO method can maintain the speed of the original classical VO system for each camera pose estimation and obtain an excellent pose estimation accuracy.

## 6. Conclusions

In this paper, we presented the stereo visual odometry pose correction network that is trained to correct classical stereo VO systems in an unsupervised manner without the need for six-DoF ground truth. We combine the multiview geometry constraints of classical stereo VO system with unsupervised learning’s ability to use unlabeled sensor data. By regressing pose corrections, the classical stereo VO system is more accurate. Our evaluation results show that the stereo visual odometry pose correction network can significantly improve the positioning accuracy of the classical stereo VO system, and the improved stereo VO system has almost reached the state of the art. Currently, our network can well correct the pose of the classical stereo VO system, but it does not optimize the map points. In the next step, we will extend our system to a visual SLAM system to optimize the map points. In the future, we also plan to incorporate other sources of metric information (e.g., inertial measurement unit data) to improve our translation corrections better.

## Figures and Tables

**Figure 1 sensors-21-04735-f001:**
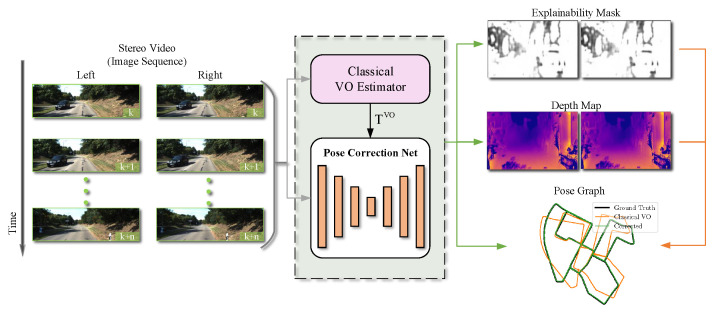
The testing framework of the proposed unsupervised stereo visual odometry pose correction network. It takes the prior pose ($T^{VO}$) produced by classical stereo VO system (e.g., ORB-SLAM2 [19], DSO [6], and LSD-SLAM [5]) and stereo color images as input and produces pose correction, depth map, and explainability mask.

**Figure 2 sensors-21-04735-f002:**
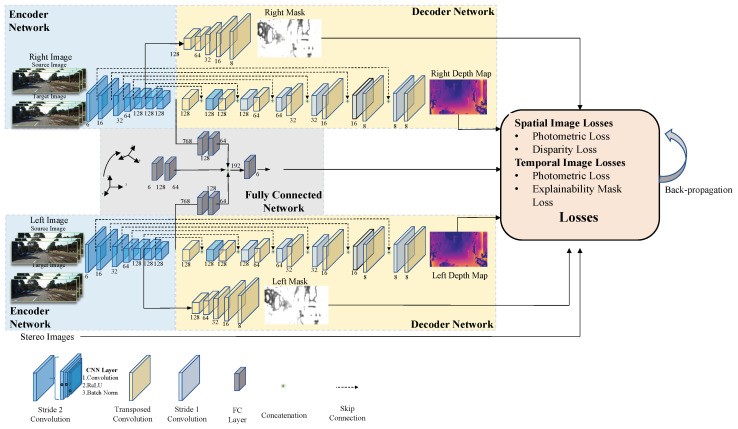
Training scheme of the proposed unsupervised stereo visual odometry pose correction network.

**Figure 3 sensors-21-04735-f003:**
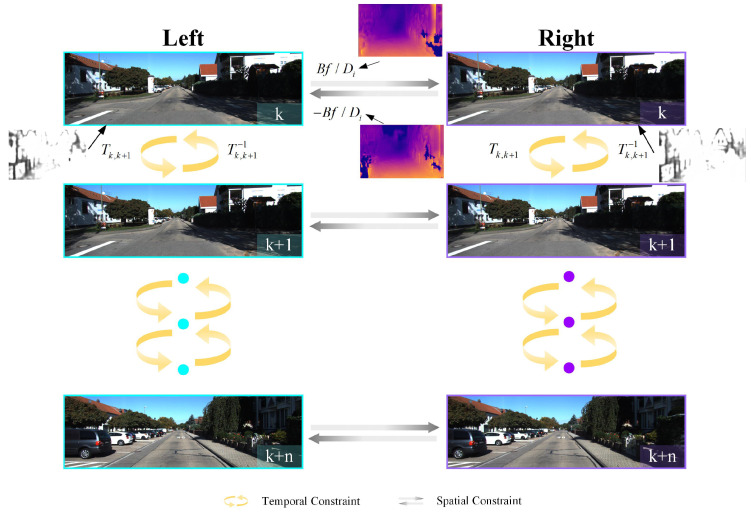
Construction of loss functions.

**Figure 4 sensors-21-04735-f004:**
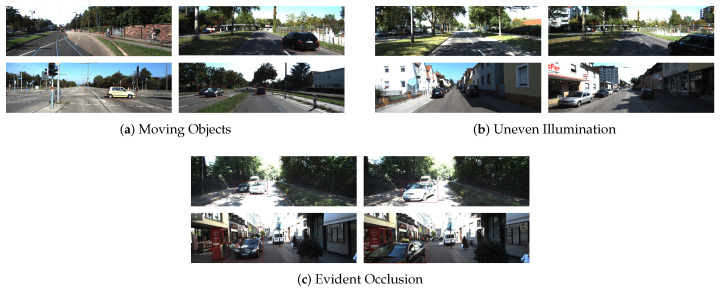
The challenging scenes in KITTI dataset: (**a**) the moving objects scene. The red boxes represent moving objects; (**b**) the uneven illumination scene; (**c**) the evident occlusion scene. The red boxes represent the object being obscured.

**Figure 5 sensors-21-04735-f005:**
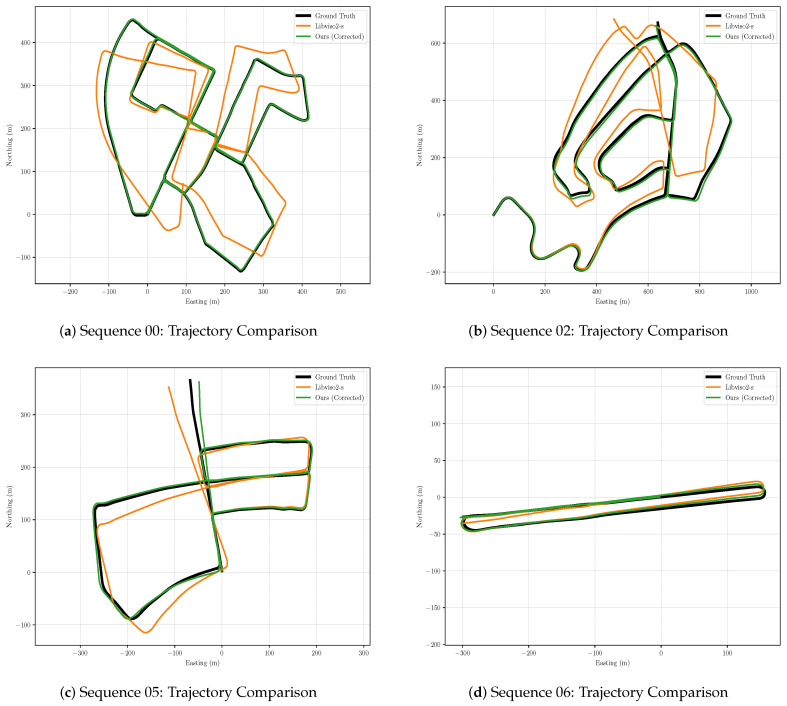
Corrected *libviso*2-*s* trajectories on KITTI sequences 00, 02, 05-10. We show the original *libviso*2-*s* estimate for comparison.

**Figure 6 sensors-21-04735-f006:**
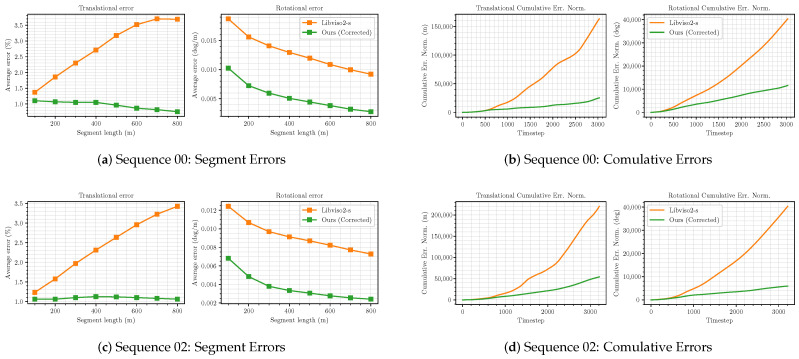
Mean segment errors and c-ATE for *libviso*2-*s* with and without pose correction net on KITTI sequences 00, 02, 05–10.

**Table 1 sensors-21-04735-t001:** Quantitative results on KITTI Odometry Sequence 00, 02, 05-10. The best result is in bold, and the second best is underlined.

Metric	Method	00	02	05	06	07	08	09	10	Avg. Err.
$t_{err}$	SfM-Learner (from [31])	21.32	24.10	12.99	15.55	12.61	10.66	11.32	15.25	15.48
	Depth-VO-Feat (from [31])	6.23	6.59	4.94	5.80	6.49	5.45	11.89	12.82	7.53
	SC-SfM-Learner (from [31])	11.01	6.74	6.70	5.36	8.29	8.11	7.64	10.74	8.07
	ss-DPC-Net	2.49	**0.93**	1.25	1.06	1.16	1.52	2.11	3.20	1.72
	ESP-VO (from [17])	-	-	3.35	7.24	3.52	-	-	9.77	5.97
	libviso2-s	2.79	2.42	2.31	1.12	3.14	2.44	2.43	1.40	2.26
	ORB-SLAM2 (from [31])	11.43	10.34	9.04	14.56	9.77	11.46	9.30	2.57	9.81
	**Ours**	**0.95**	1.08	**1.06**	**0.73**	**0.84**	**1.42**	**1.70**	**0.89**	**1.08**
$r_{err}$	SfM-Learner (from [31])	6.19	4.18	4.66	5.58	6.31	3.75	4.07	4.06	4.85
	Depth-VO-Feat (from [31])	2.44	2.26	2.34	2.06	3.56	2.39	3.60	3.41	2.76
	SC-SfM-Learner (from [31])	3.39	1.96	2.38	1.65	4.53	2.61	2.19	4.58	2.91
	ss-DPC-Net	1.41	0.42	0.45	0.54	0.95	0.80	0.80	1.19	0.82
	ESP-VO (from [17])	-	-	4.93	7.29	5.02	-	-	10.2	6.86
	libviso2-s	1.29	0.92	1.13	0.79	1.68	1.39	1.19	1.06	1.18
	ORB-SLAM2 (from [31])	0.58	**0.26**	**0.26**	**0.26**	**0.36**	**0.28**	**0.26**	**0.32**	**0.28**
	**Ours**	**0.53**	0.37	0.44	0.41	0.88	0.75	0.58	0.43	0.54
ATE	SfM-Learner (from [31])	104.87	185.43	60.89	52.19	20.12	30.97	26.93	24.09	63.19
	Depth-VO-Feat (from [31])	64.45	85.13	22.15	14.31	15.35	29.53	52.12	24.70	38.47
	SC-SfM-Learner (from [31])	93.04	70.37	40.56	12.56	21.01	56.15	15.02	20.19	41.11
	ss-DPC-Net	15.16	36.87	8.20	6.96	5.88	42.32	28.91	30.75	21.88
	libviso2-s	64.42	84.61	25.02	7.71	14.47	65.68	48.88	9.46	40.03
	DSO (from [46])	113.18	116.81	47.46	55.62	16.72	111.08	52.23	11.09	65.52
	ORB-SLAM2 (from [31])	40.65	47.82	29.95	40.82	16.04	43.09	38.77	5.42	32.82
	CNN-SVO (from [46])	17.53	50.52	8.15	11.51	6.51	**10.98**	**10.69**	4.84	15.09
	**Ours**	**10.47**	**19.38**	**7.00**	**4.40**	**3.28**	37.48	24.90	**3.28**	**13.77**
RPE(M)	SfM-Learner (from [31])	0.282	0.365	0.158	0.151	0.081	0.122	0.103	0.118	0.173
	Depth-VO-Feat (from [31])	0.084	0.087	0.077	0.079	0.081	0.084	0.164	0.159	0.102
	SC-SfM-Learner (from [31])	0.139	0.092	0.070	0.069	0.075	0.085	0.095	0.105	0.091
	ss-DPC-Net	0.050	0.063	0.037	0.047	0.037	0.051	0.054	0.042	0.048
	libviso2-s	0.062	0.078	0.051	0.063	0.052	0.068	0.069	0.059	0.062
	ORB-SLAM2 (from [31])	0.169	0.172	0.140	0.237	0.105	0.192	0.128	0.045	0.315
	**Ours**	**0.041**	**0.053**	**0.031**	**0.042**	**0.031**	**0.047**	**0.049**	**0.038**	**0.041**
RPE${(}^{\circ })$	SfM-Learner (from [31])	0.227	0.172	0.153	0.119	0.181	0.152	0.159	0.171	0.167
	Depth-VO-Feat (from [31])	0.202	0.177	0.156	0.131	0.176	0.180	0.233	0.246	0.188
	SC-SfM-Learner (from [31])	0.129	0.087	0.069	0.066	0.074	0.074	0.102	0.107	0.089
	ss-DPC-Net	0.095	0.082	0.076	0.073	0.084	0.083	0.067	0.077	0.080
	libviso2-s	0.117	0.095	0.091	0.093	0.077	0.095	0.086	0.093	0.092
	ORB-SLAM2 (from [31])	**0.079**	0.074	**0.058**	**0.055**	**0.047**	**0.061**	**0.061**	**0.065**	**0.062**
	**Ours**	0.093	**0.072**	0.066	0.063	0.054	0.071	0.063	0.071	0.069

## Data Availability

Not applicable.

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
