# Peer review of "Stereo Visual Odometry Pose Correction through Unsupervised Deep Learning"

_sensors, 2021, doi:10.3390/s21144735_

Round 1
Reviewer 1 Report
The Authors have introduced an unsupervised deep learning approach, which provides a pose correction for the stereo visual odometry. The proposed system was evaluated during extensive experiments. The contribution of this paper is widely discussed in the context of related works.
The benefits of the proposed approach should be better exposed in the paper. The Authors have concluded that “the improved stereo VO system has almost reached the state-of-the-art”. Especially, the ORB-SLAM2 method has provided better results than the proposed one (in terms of rotational error and relative pose error). These facts suggest that the existing methods are at least as good as the approach introduced in this paper. Therefore, additional explanations are necessary to clarify the advantages of the proposed approach in the context of the state-of-the-art.
The schema shown in Fig. 1 should be better explained in the text. For instance, the meaning of the symbol T^vo as well as the meaning of the green/orange trajectory, is not clear.
The dataset used in experiments should be better described in Section 5. Does the dataset include sequences with illumination changes, dynamic scenes, and occlusion? Please describe the “training pairs”.
Please consider using charts for better visualization of the results (average errors) in Tab. 1.
Author Response
Dear Reviewer:
Thank you for your letter and for the comments concerning our manuscript entitled “Stereo Visual Odometry Pose Correction through Unsupervised Deep Learning” (ID: sensors-1284455). Those comments are all valuable and very helpful for revising and improving our paper, as well as the important guiding significance to our researches. We have studied comments carefully and have made correction which we hope meet with approval. The main correction in the paper and the responds to your comments are in the attachment.
Once again, thank you very much for your comments and suggestions.

Reviewer 2 Report
The paper presents an unsupervised deep learning-based method for visual odometry which is used to correct pose estimation provided by the "classical VO estimator". I find the paper to fit nicely with the journal scope, as well as having the potential to be interesting to the journal audience. It is written in a concise manner and presents results that support made claims. Once department where some additional improvements are needed (and thus my recommendation) is the presentation: some information is missing and some additional data should be included.
More specifically my comments/suggestions are as follows (in order of appearance and not importance - this should be deduced by the authors):
- I would suggest augmenting the abstract with some results numbers to better illustrate the algorithm's performance and be self-contained.
- the use of the English language is fine and appropriate. However, there are still some minor grammar-related issues throughout the paper (e.g. "Recently, researchers use deep learning (DL) based.....") which should be corrected. Thus, I would suggest carefully proofreading the paper or putting it through some automatic language correction system, like Grammarly.
- Figure 1 - classical VO Estimation is mentioned in the figure but more information (e.g. which one) would be provided here and in other appropriate places throughout the paper.
- all abbreviations should be defined the first time they are used (no matter how well they are known). This is currently not the case for all abbreviations (e.g. ORB). Please find and correct such issues.
- "The main reason for restricting the development of the supervised VO system is that it requires many labeled data to train the network" - please do not use vague terms (here and in the rest of the paper) like "many". Please provide a concrete number or at least an order of magnitude.
- it would be nice if authors positioned their work within the division they introduced: Geometry-based, supervised learning, unsupervised or hybrid. This is because based on Figure 1 it seems (at least to me) to be in the real of the hybrid approach.
- "The network inputs are two pairs of images (source image and target image)...." - however in Figure 2 to which this sentence pertains there are no par of images labels as "target" nor "source". Please unify your terminology in the text and figures.
- Figure 2 - some of the numbers used to depict NN layer sizes are too hard to read (too small font). Please try increasing the font size a bit. Also in the middle part of the figure, there seems to be a coordinate frame and its rotation, but no axis labels are present. Please be as specific as possible in the figures too.
- please name and explain all parameters/variables used in the equations. This is done for most of them, but for example for equation (1) this is not the case - what do sub/super-scripts mean? It might be self-evident but still needs explicit definition to avoid any ambiguity.
- "The whole network consists of encoder network, decoder network, and fully connected network." - clearly marking these main parts of the network in figure 2 might be a nice addition that should increase understanding of the figure.
- I would suggest including some introductory text between section 4.2. and sub-section 4.2.1.
- results are very detailed and compare the method with many state-of-the-art methods. However, what I'm missing from the results is computational complexity analysis of the proposed method and can it run in real-time (at what rate?). Comparison (in terms of computational complexity) with other methods is not needed (but a comment to that end would be welcomed).
- how were hyperparameters alpha_1, alpha_2, alpha_3, and alpha_4 determined/selected?
- a more explicit and compelling explanation as to why particular sequences from the KITTI datasets were chosen should be provided in the text. Also providing some example images from the database could be informative to the reader.
- a more explicit definition of the ground truth used in the database and in the work should be provided.
- Figure 4 - I would suggest rearranging legends in particular sub-figures so that they do not overlap with the trajectory (this makes images unnecessarily hard to "read")
- it is a bit unclear how large was the dataset used for training the network and how large was used for testing? Are reported values based on the testing set? This information should be clearly stated in the manuscript in the appropriate sub-sections.
- Table 1 - table one is a bit hard to "use" i.e. compare particular values. One needs to skip 4 rows to get to the next comparable items. Thus I would urge the authors to think of some alternative, easier to "use" table format.
- I would suggest including detected drawbacks/limitations of the proposed method in the conclusion section, alongside ways/suggestions on how to deal with them (In the future).
- there are some nice references (some of them from MDPI) that use very similar systems and testing procedures/datasets like "Unsupervised Deep Learning-Based RGB-D Visual Odometry" from Liu et. al. published in Applied sciences (Appl. Sci. 2020, 10, 5426). I would urge the authors to consider including them in the introduction/related works section. This is just a suggestion.
Author Response
Dear Reviewer:
Thank you for your letter and for the comments concerning our manuscript entitled “Stereo Visual Odometry Pose Correction through Unsupervised Deep Learning” (ID: sensors-1284455). Those comments are all valuable and very helpful for revising and improving our paper, as well as the important guiding significance to our researches. We have studied comments carefully and have made correction which we hope meet with approval. The main correction in the paper and the responds to your comments are in the attachment.
Once again, thank you very much for your comments and suggestions.

This manuscript is a resubmission of an earlier submission. The following is a list of the peer review reports and author responses from that submission.